# Empowering the Internet of Things Using Light Communication and Distributed Edge Computing

Abdelhamied A. Ateya [1,*], Mona Mahmoud [1], Adel Zaghloul [1], Naglaa. F. Soliman [2] and Ammar Muthanna [3,4]

1 Department of Electronics and Communications Engineering, Zagazig University, Zagazig 44519, Egypt; Mona.saleh@eng.zu.edu.eg (M.M.); AZElsayed@eng.zu.edu.eg (A.Z.)
2 Department of Information Technology, College of Computer and Information Sciences, Princess Nourah bint Abdulrahman University, P.O. Box 84428, Riyadh 11671, Saudi Arabia; nfsoliman@pnu.edu.sa
3 Department of Telecommunication Networks and Data Transmission, The Bonch-Bruevich Saint-Petersburg State University of Telecommunications, 193232 Saint Petersburg, Russia; muthanna.asa@spbgut.ru
4 Department of Applied Probability and Informatics, Peoples' Friendship University of Russia (RUDN University), 117198 Moscow, Russia
* Correspondence: a_ashraf@zu.edu.eg; Tel.: +2-0100-523-7673

**Abstract:** With the rapid growth of connected devices, new issues emerge, which will be addressed by boosting capacity, improving energy efficiency, spectrum usage, and cost, besides offering improved scalability to handle the growing number of linked devices. This can be achieved by introducing new technologies to the traditional Internet of Things (IoT) networks. Visible light communication (VLC) is a promising technology that enables bidirectional transmission over the visible light spectrum achieving many benefits, including ultra-high data rate, ultra-low latency, high spectral efficiency, and ultra-high reliability. Light Fidelity (LiFi) is a form of VLC that represents an efficient solution for many IoT applications and use cases, including indoor and outdoor applications. Distributed edge computing is another technology that can assist communications in IoT networks and enable the dense deployment of IoT devices. To this end, this work considers designing a general framework for IoT networks using LiFi and a distributed edge computing scheme. It aims to enable dense deployment, increase reliability and availability, and reduce the communication latency of IoT networks. To meet the demands, the proposed architecture makes use of MEC and fog computing. For dense deployment situations, a proof-of-concept of the created model is presented. The LiFi-integrated fog-MEC model is tested in a variety of conditions, and the findings show that the model is efficient.

**Keywords:** internet of things; multiple access edge computing; light fidelity; fog computing; latency

## 1. Introduction

The Internet of Things (IoT) is a recent communication system that considers connecting heterogeneous distributed wireless devices and provides the communication medium for the interaction among remote machines [1]. It enables the communication paradigm of machine-to-machine (M2M), introduced to enable the interactions among machines, e.g., sensors and actuators [2]. Internet-connected devices went from 5 million to billions in just one year [3]. It is estimated that the number of connected devices per kilometer in 2025 will be 20 times higher than the current existing number, creating a revenue increase of 10 times as many as the current [3].

The IoT has provided the inanimate physical environment with a digital nervous system. The IoT has been widespread recently, revealing its capabilities in various applications from automated automobiles and wearables to smart homes and cities, generating a global influence [4]. More than 25 billion devices are currently connected to the IoT, generating a flood of information that must be monitored and analyzed [5]. As a result, they learn and

improve from the accessible data sets without any intervention. That is how IoT gadgets become smart.

Connectivity is a vital component in IoT projects. Stakeholders must determine the most appropriate connectivity model for their projects and products. With more than 30 IoT connectivity choices being commercially available, the choice can be challenging, considering their ongoing development [6]. Heterogeneous IoT networks assign either short-range or long-range communication technologies. However, such communication interfaces do not meet the requirements of IoT networks, including energy, throughput, reliability, availability, and latency [7].

Visible light communication (VLC) is a promising technology that can be introduced for IoT networks to overcome the previously mentioned challenges [8]. Light Fidelity (LiFi) is a revolutionary technology that enables vast volumes of data to be transmitted quickly. It is a form of VLC, which provides an efficient medium for achieving ultra-high reliability and availability, low latency, and higher energy efficiency for IoT applications [9,10]. The LiFi industry has already evolved into a fiercely competitive market, with several large enterprises and industry leaders competing for market share. Many market solutions are available with the appropriate size and integration [11]. This facilitates the implementation of this technology in IoT networks.

Moving from centralized cloud computing schemes to distributed computing paradigms achieves many benefits to the communication networks, including the reduction of communication latency, the introduction of novel services, increased spectral efficiency, increased network availability, and reduced congestion of the core network. Distributed edge techniques deploy distributed servers at the edge of the access network instead of a remote single centralized one that should have an interface via the core network. Such distributed servers have interfaces to the remote centralized cloud [12].

Limited computing resources of IoT devices is a challenge that can be managed using distributed edge computing technology, e.g., fog computing and multiple access edge computing (MEC). With a distributed cloud model, enterprises reduce network congestion, latency, and danger of data loss. Furthermore, enterprises can better assure consistency of data integrity requirements because data can be stored in the nation it is produced. The distributed cloud is a way of having storage, accounts, and networks in a small cloud that exists outside the primary cloud [13]. Establishing a distributed cloud brings computers closer to the end-user, enabling distributed cloud processed data in real-time, lower response time, and chances for better security. Instances of distributed clouds are edge computing and fog computing. MEC is an architecture standard for edge computing, while fog computing is an umbrella suite for edge computing [14].

This work considers designing and developing a general framework for IoT networks using LiFi and a distributed edge computing scheme. It aims to enable dense deployment, increase reliability and availability, and reduce the communication latency of IoT networks. A multi-level distributed edge paradigm based on a fog-MEC model is introduced. For dense deployment situations, a proof-of-concept of the created model is presented. The main contributions of the work are summarized as follows.

1. Designing and developing a LiFi-based IoT network.
2. Designing and developing a fog-MEC model to provide computing resources at the edge of the LiFi-based IoT networks.
3. Performance evaluation of the developed LiFi-based IoT model for multiple scenarios.

The article is organized as follows. We will go through some background information and related works in Section 2. Section 3 outlines the proposed fog-MEC IoT network based on LiFi. Section 4 presents the outcomes of the assessment and discusses the findings in order to validate the generated model. Conclusions and recommendations for the future are discussed in Section 5.

## 2. Background and Related Work

This work considers two main technologies to overcome the previously introduced challenges and develop a reliable IoT framework: LiFi technology and distributed edge computing. This section considers introducing a background on both technologies and the existing related works that consider such technologies for IoT networks.

### 2.1. LiFi for IoT Networks

LiFi addresses indoor wireless access as the first use case; however, with recent advances in communication systems and the current demands, many new use cases are introduced [9]. Among these use cases, IoT represents a promising one. The IEEE 802.11bb standard group mentioned eight envisaged use cases and classified their associated environments, applications, and data traffic conditions [15]. Industrial wireless, wireless access in medical environments, enterprise networks, and secure home networks were identified as primary use cases [16]. Moreover, the standard group defined the secondary use cases including vehicle-to-vehicle (V2V) communications, communication between vehicles and roadside units, i.e., vehicle-t-infrastructure (V2I) communications, backhaul communications, and underwear communications [15].

Such primary and secondary use cases mainly deploy the IoT paradigm, and LiFi can be introduced to assist the communication of IoT-based networks. LiFi is an efficient solution for many IoT applications and use cases, including indoor and outdoor applications. We can list a part of such applications in the following points [17–19].

- Industrial IoT (IIoT): IIoT is an ultra-reliable low latency communication (uRLLC) that requires a very low latency with ultra-high system availability and reliability. This can be achieved by introducing visible light communications with the distributed edge paradigm to these systems. LiFi can be used to achieve reliable network connectivity with the required ultra-low end-to-end latency.
- Medical IoT (IoMT): LiFi can be used to provide the required coverage of IoMT networks with the required ultra-high system availability and reliability since reliability and availability are major challenges with such networks. Moreover, the required coverage area of such networks makes LiFi technology convenient for such systems.
- Underwater IoT (IoUT): Underwater communication faces many challenges associated with the propagation of radio waves through the communication medium. LiFi can achieve a coverage distance six times that of radio waves, making it more convenient for IoUT applications.
- Vehicular IoT (IoVT): LiFi can provide an efficient channel for V2V applications with ultra-high reliability. Introducing LiFi to IoVT reduces the communication overhead and achieves higher latency efficiency. Furthermore, LiFi can be used to offload data between vehicles and roadside units (RSUs).

There are many advantages of introducing LiFi technology to IoT networks, which can be summarized as follows [20–23].

1. Signaling: With IoT-connected devices, dependable bidirectional signaling is critical for convenient data routing. Li-Fi offers a highly reliable data rate of up to 10 Gbps.
2. Security: When delivering or obtaining a data stream, lights cannot pass through walls and doors. This increases security and control over who can connect.
3. Spectrum usage: Wireless devices have a massive untapped pool of resources because the light beam is 1000 times wider than the full 300 GHz radio, microwave, and millimeter-wave radio spectrum. As a result, enough capacity is available to support a high number of IoT devices.
4. Omnipresent detection: It can detect when an IoT system disconnects from or reconnects to the network in real-time. Li-Fi can detect IoT devices and resolve any network issues. As a result, Li-Fi boosts the IoT network stability.

5. Power consumption: Because LEDs are low-power gadgets, Li-Fi has significantly low power usage. Therefore, it consumes less energy than WiFi and is more environmentally friendly.
6. Massive machine communication (MMC): Massive MIMO systems that operate in the visible light range have large bandwidth.
7. LiFi everywhere: Li-Fi can be implemented and used in all indoor locations, it is human friendly, generates less interference between devices, and has a low deploying cost.

However, there are some disadvantages of introducing LiFi to IoT networks, which include the following [20–23].

1. Due to the shorter distance between the transmitter and receiver, the signal-to-noise ratio (SNR) is exceptionally high.
2. Li-Fi can only be used on devices with a LiFi sensor.
3. Direct line-of-sight (LOS) between the sender and receiver is essential for life.
4. It is less efficient for outdoor applications due to limited coverage area.

Many proposals consider developing IoT networks using LiFi technology. In ref. [24], the authors suggested a new multi-tier waveform backed by a conceptual framework and experimental studies. A universal waveform is provided by design. The LiFi-based multi-tier waveform has multi-service characteristics and can be used globally. It provides both low- and high streams that various receivers can collect. It also includes inbuilt beacons for locating and dimming control.

In ref. [25], the authors combined LiFi with the IoT to compensate for the lack of radio frequency capacity, establishing new communication channels using available systems. According to the authors, real-time image analysis can be utilized for various applications, including monitoring systems, medical image processing, machine vision, and traffic monitoring, which employs an embedded CPU loaded with a real-time operating system (RTOS). The work focused on a single kind of IoT application; however, our proposed work is entirely different.

In ref. [26], the authors offered simulation software that enables horizontal handover in a LiFi indoor network. An algorithm for realizing users' movement across neighboring cells based on a priority mechanism was implemented in this framework. According to simulation results, the suggested prioritization system enhances the quality of service (QoS) for high-speed users. The work completely differs from our proposed work since we consider LiFi for IoT applications with network modeling and performance evaluation.

In ref. [27], the authors demonstrated that LiFi is a viable solution for bringing affordable wireless connectivity to remote areas where light can reach. Because LiFi gives the best speed compared to regular WiFi, the range of throughput that LiFi will attain shows that LiFi is the ideal tool in IoT networks. This work presents a complete conceptualization of enabling the IoT using LiFi technology. It differs from our work, which visualizes the architecture that demonstrates this technology using the IoT.

In ref. [28], the authors developed a Human Safety Measurement (HSM) technique for deep mines using innovative LiFi technology. The proposed technique would have handled the difficulty of avoiding environmental risks of the undermining operation using a LiFi communication channel. With a real-time testbed setup, their technique was practically proven. The end-users receive alert messages as a result of the calculated value. The structure monitors sound levels and air changes and then sends updates to an online server via the item speak platform, which runs on a network edge. It was carried out by leveraging LiFi technology to solve connectivity issues in the mining process.

In ref. [29], the authors demonstrated how LiFi technology might be used to safely exchange acquired IoT data via the cloud. It explains how to resolve various cloud-related concerns and encourages customers to save their data on a cloud storage service. The offered software system ensures that the data saved in the cloud is secure for the enterprise.

In ref. [30], the authors examined the idea of combining IoT with LiFi and the obstacles and prospects of the combined system for developing unique and intelligent solutions. There are no further illustrations and network modelling.

In ref. [23], the authors presented an overview of the implementation of the Visible Light Communication system, emphasizing its potential, the future of the IoT, and its challenges. With the expected growth rates of IoT deployment, it can be considered a success. LiFi is a feasible technology for meeting the IoT framework deployment's substantial bandwidth requirements. This work differs from ours, as we presented a complete model of the IoT with LiFi, clarifying some indoor and outdoor applications.

In ref. [31], the authors deployed LiFi technology to assist IoT communications in bidirectional transmission. The non-orthogonal multiple access (NOMA) scheme was used to increase the energy efficiency of the network. The maximum energy was achieved by obtaining the optimal power allocation model. The work investigated the energy efficiency of LiFi-based IoT using different multiple access schemes; however, results demonstrated that NOMA with the optimal power allocation improved the energy efficiency.

In ref. [32], the authors introduced a novel LiFi-based IoT network structure. The authors deployed LiFi for IoT networks instead of radiofrequency (RF) spectrums, e.g., WiFi, Zigbee, and Bluetooth, due to spectrum limitations of such RF bands. LiFi is installed in a distributed environment in the system, where data should be collected. The collected Li-Fi data is forwarded and analyzed by the IoT network. The work mainly presents a general survey of LiFi devices and presents the IoT network structure based on LiFi technology; however, there are no further illustrations and network modelling. Moreover, there is no evaluation of the work, and thus, the work completely differs from our proposed work.

### 2.2. Distributed Edge Computing for IoT Networks

Edge computing and IoT are linked together because IoT devices usually do not have computing power internally and depend on cloud resources. Commercially, there are two primary types of edge computing: fog computing and MEC computing [33]. Fog computing is a new distributed computing paradigm that was just introduced with limited computational power and greater flexibility in deployment. In comparison to MEC units, fog nodes have fewer resources, but they are more adaptable [34].

MEC implementations can ensure a transmission latency of milliseconds or less in most cases. They can also do real-time data processing, reducing the frequency and amount of data transmissions to any central point. Sensitive information can also be restricted to local areas, improving the firm's security [35].

Fog computing is an excellent fit for IoT applications that create terabytes of data, requiring a substantial amount of information processing and transporting data back and forth to the cloud [36]. Fog computing can be quite valuable in a variety of IoT applications. MEC and fog concepts for IoT networks are examined in this section of the literature.

In ref. [37], the authors developed an air-MEC system to assist IoT networks. Unmanned aerial vehicles were deployed to support IoT end devices with MEC services. A swarm of UAVs were considered for the developed system, and the offloading problem was modeled. A resource scheduling algorithm based on deep reinforcement was developed to achieve higher performance. The developed algorithm balances the load among the deployed UAVs.

In ref. [38], the authors considered the energy consumption of fog-based IoT networks. The work deployed the paradigm of green energy to improve the energy performance of fog-IoT networks. A genetic algorithm was introduced to enable accepting the maximum number of tasks at the required service quality. The key performance metrics considered in the evaluation process were the delay and the energy consumption.

In ref. [39], the authors developed a general framework to integrate IoT with fifth-generation cellular (5G) systems using distributed edge computing. Fog and MEC paradigms were used for IoT and 5G cellular systems, and the platform provided interfaces between edge servers. Moreover, the work provided an energy-aware and latency efficient offload-

ing scheme for the developed framework. The core network of the systems was managed by software-defined networking (SDN) of multiple controller schemes. The system was evaluated for heterogeneous scenarios, and the developed model was validated over a developed IoT testbed of a large number of end devices.

In ref. [40], the authors developed an offloading scheme of three layers to assist MEC-based networks. The three considered offloading layers were the end devices, cloudlet, and the remoted cloud. Tasks with excessive communication costs are handled locally at the end device; however, tasks with excessive computing costs are handled at the edge of the remote cloud layers. In the greedy-based offloading scheme, the scheduling is aided by the computing capabilities of the device, with a greedy optimization strategy used to reduce the task communication cost.

The novelty of this work comes from introducing LiFi as the communication interface for IoT networks and testing this for IoT networks. Moreover, the deployment of two levels of distributed edge computing for IoT networks is novel. Fog-MEC model integration and proof-of-concept for LiFi-based IoT networks with dense deployment are being considered for the first time, to the best of our knowledge.

## 3. Proposed LiFi-Based IoT Framework

In this section, we provide a general framework for LiFi-based IoT networks. The considered IoT networks deploy a distributed edge computing paradigm with a novel fog-MEC model. Figure 1 introduce the end-to-end structure of the developed framework.

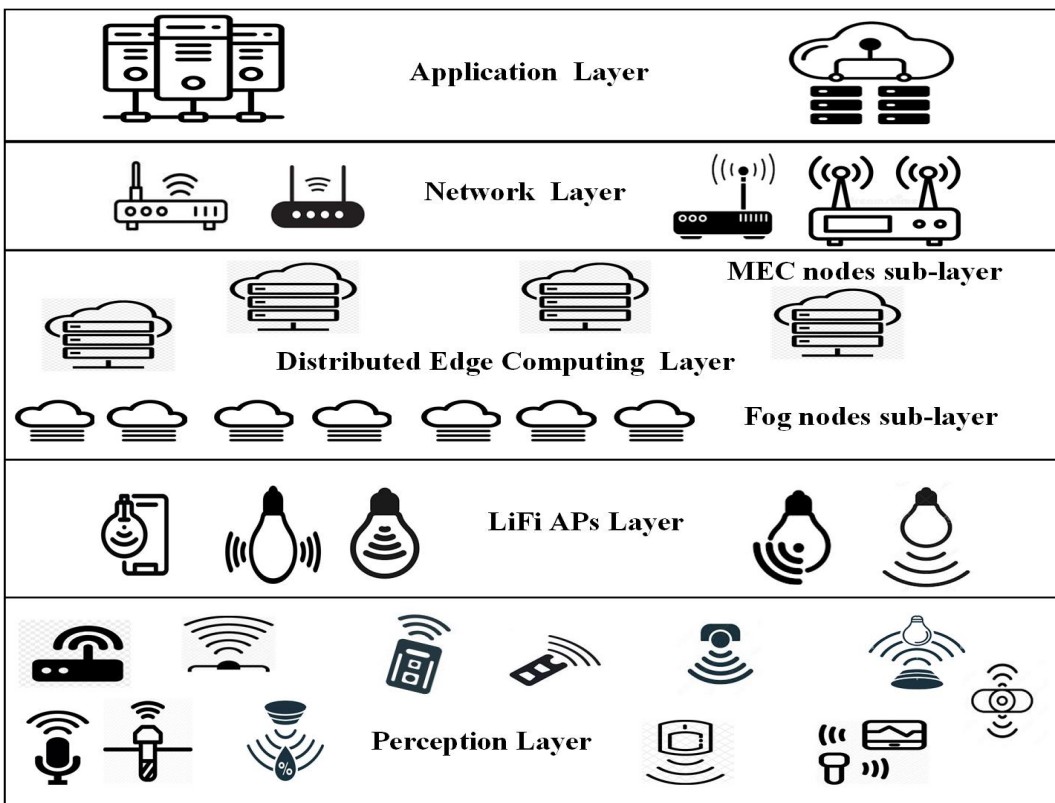

**Figure 1.** Layering system of the developed LiFi-based IoT system.

The developed LiFi-based IoT system consists of the four layers introduced in Figure 1. The system deploys two main layers over the IoT systems' traditional layers: the LiFi communication layer and distributed edge layer.

### 3.1. LiFi-Communications for IoT Network

The tier is in charge of developing the methods of communication between the source of data and the data storage hub. LiFi can be installed in simple LED bulbs to send signal information through the light. The LiFi bulbs serve as the communication's endpoint, and they can be attached to the LAN via standard internet architecture (e.g., Ethernet) or with the help of other wireless systems.

The LiFi communication layer deploys distributed access points to enable visible light communications to IoT end devices. LiFi access points should be oriented to reduce the overlapped regions between coverage areas and achieve the required full coverage. However, this introduces an intra interference, i.e., interference between access points, that should be managed.

Another challenge with the considered LiFi communications is that the handover process takes place when the end device moves between the coverage of the two neighboring LiFi access points. This issue is critical in LiFi communications, especially with the mobility of end devices. Moving between LiFi access points increases communication overhead and latency which affect the overall system performance. Thus, this issue should be managed in a way that does not affect the required performance.

A hierarchal structure of the LiFi access points is used to overcome the previously introduced challenges. Each group of LiFi access points can form a multiple-input multiple-output (MIMO) network with a fronthaul connection to a central point, i.e., a LiFi controller. Figure 2 present the considered hierarchal structure of a LiFi-based network.

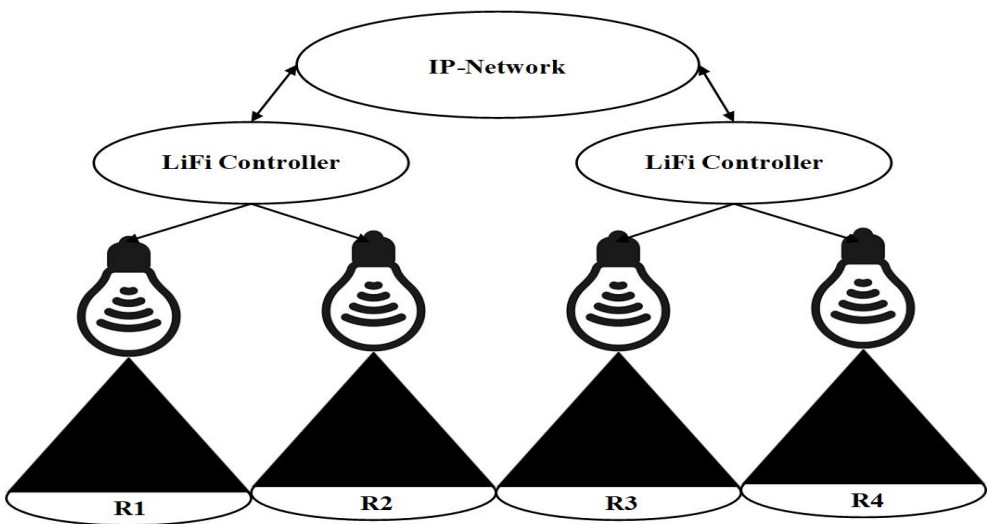

**Figure 2.** Hierarchal structure of the considered LiFi system.

Devices belonging to the same group of access points communicate via none orthogonal multiple access (NOMA) transmissions [31]. Multiple end-users can transmit simultaneously using NOMA. Moreover, the light propagation model introduced in [31] is used. The uplink and downlink communications are established by altering the light intensity of the LED via a driver module. The LiFi controller has an interface to the IoT gateway connected to the MEC unit. The developed model assumes end devices with hybrid motes to support LiFi and radio waves communications.

### 3.2. Fog-MEC Model for IoT Network

The proposed IoT paradigm features a four-layer structure, as shown in Figure 1. Two tiers of edge computing units are used in the created distributed edge computing-based IoT network. Fog computing units are installed between end devices and edge gateways at this initial level. The detected data is pre-processed and computed using distributed fog units, which are placed close to the end devices. LiFi access points may be linked to these

devices, or they can be installed as stand-alone nodes. These nodes serve as a means of transferring data to and from end devices that are located close by.

MEC servers are the second tier of edge computing devices linked to IoT access points and gateways. Introducing MEC units provides a second path for data offloading that reduces the core network congestion, increases the overall network scalability and reliability, and provides a second level of computing resources and energy assistance to end devices. Artificial intelligence (AI) algorithms may now be implemented in MEC devices, allowing for the necessary level of network automation.

Fog and MEC units can both host and support services in a matured edge computing paradigm depicted in Figure 3. Our previously developed offloading scheme presented in [41,42] is considered for the developed two-tire edge computing-based IoT system. Fog nodes or MEC servers can be used to do tasks that are too complex or of high energy cost. Figure 4 depict the levels of offloading of the created fog-MEC scheme, and Algorithm 1 is the considered offloading method for the developed model.

---

**Algorithm 1.** Energy and Latency-Aware Offloading Algorithm for Fog-MEC Model

---

1:  Initialize QoS parameters and Energy threshold of the device (IoT end device/fog/MEC)
2:  Calculate task specification parameters using the program profiler
3:  Calculate the local execution time
4:  **If** (local execution time meets QoS)
5:  Calculate the energy required for local handling of the task
6:  **If** (remaining energy after task execution > energy threshold level of the device)
7:  Handle the task locally
8:  **end if**
9:  **else**
10: Request offloading of the higher level
11: Process offloading request
12: **If** (Time and energy decisions for accepting offloading are positive)
13: Accept offloading request
14: Offload the task to the dedicated server
15: Handle the task
16: Send result
17: **else**
18: Reject offloading request
19: Terminate the task
20: **end if**
21: **end if**

---

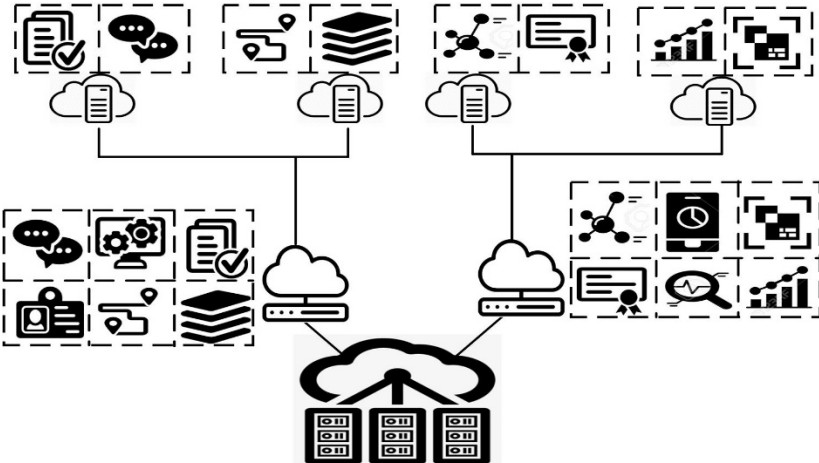

**Figure 3.** Main levels of the developed fog-MEC scheme.

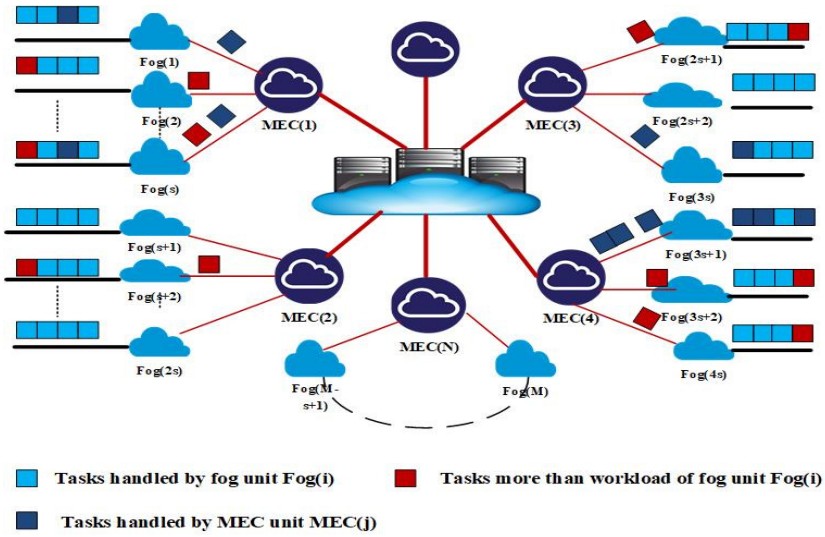

**Figure 4.** Levels of offloading of the proposed fog-MEC scheme.

The IoT end device's decision engine chooses whether to process the job locally or send it to a fog or MEC node based on the current condition of the resources available there. IoT end devices offload their computational chores to fog nodes as the first offloading choice if the local execution fails. Fog nodes respond to requests for offloading based on the current state of their available resources and the required QoS for each task. The MEC server attached to the serving gateway takes over tasks that fog nodes are unable to handle. For network applications, machine learning algorithms can be executed on fog or MEC units. Fog nodes are used for lightweight algorithms; MEC units, on the other hand, should be used for more complex ones.

Using the multi-server queuing model M/M/s, the average response time of the computing server can be calculated based on the arrival rate of tasks. The average response time of the fog server is calculated using the Erlang-C formula as follows.

$$T_{Q-fog-i}(\lambda) = \frac{E_C\left(s_i, \frac{\lambda_i}{\mu_i}\right)}{s_i\mu_i - \lambda_i} + \frac{1}{\mu_i}, \tag{1}$$

$$E_C(n, \varphi) = \frac{\left(\frac{(s\varphi)^n}{n!}\right)\left(\frac{1}{1-\varphi}\right)}{\sum_{k=0}^{n-1}\frac{(n\varphi)^k}{k!} + \left(\frac{(n\varphi)^n}{n!}\right)\left(\frac{1}{1-\varphi}\right)}, \tag{2}$$

where $T_{Q\text{-}fog\text{-}i}$ is the average response time of the $i$th fog node that has S servers, $\lambda$ is the arrival rate of tasks, and $\mu_i$ is the service rate. Similarly, the average response time of the MEC server, $T_{Q\text{-}MEC\text{-}i}$, can be calculated as follows.

$$T_{Q-MEC-i}(\lambda) = \frac{E_C\left(s_i, \frac{\lambda_i}{\mu_i}\right)}{s_i\mu_i - \lambda_i} + \frac{1}{\mu_i}, \tag{3}$$

## 4. Performance Evaluation

The proposed system, including the LiFi-IoT with the fog-MEC model, is evaluated for heterogeneous scenarios over reliable environments. The considered performance metrics are latency, availability, reliability, and resource utilization. The considered latency is the time required to handle a computing task, including offloading latency. Availability and reliability are indicated by measuring the communication overhead and the percentage of blocked tasks.

### 4.1. Simulation Setup

The developed LiFi-based IoT system was evaluated using the NS-3 platform and the modified CloudSim environment introduced in [41]. NS-3 is a trusted and trustworthy simulation environment for creating networking protocols that can be implemented in real-world networks. We adopted the libraries introduced in [43] for the developed system.

A network of 12 LiFi access points and 30 randomly distributed end devices was considered. Four LiFi access points were considered controllers with interfaces to IoT gateways. The considered simulation parameters are introduced in Table 1. NOMA was used for uplink and downlink transmission, with the propagation model introduced in [31].

**Table 1.** Simulation parameters.

| Parameter | Value |
|---|---|
| Number of LiFi-Access points | 12 |
| Number of LiFi controllers | 4 |
| Number of end devices | 10, 20, 30, 40, 50, 60, 70 |
| Network area | $16 \times 12 \text{ m}^2$ |
| LED half-power semi angle | $70°$ |
| Reflectivity factor | 0.8 |
| Transmission power | 8.8 w |
| Refractive index | 1.5 |
| Maximum vertical distance | 2.5 m |
| Minimum vertical distance | 1.5 m |
| Maximum horizontal distance | 3 m |
| Minimum horizontal distance | 0 m |
| Active area of photodetector | $1 \text{ cm}^2$ |
| Photodetector responsivity | 0.5 A/W |
| Receiver half-angle | $70°$ |
| Optical filter gain | 1 |
| Noise Power spectral density | $10^{-22} \text{ A}^2/\text{Hz}$ |
| Bandwidth | 20 MHz |
| Arrival rate ($\lambda$) | 15 |
| Maximum workload (fog) ($W_{\text{max-fog}}$) | 30 (event/s) |
| Maximum workload (MEC) ($W_{\text{max-MEC}}$) | 100 (event/s) |
| Fog node | |
| Storage/RAM | 512 Mb |
| Processing/CPU | $\ni [0.1, 0.3]$ GHz |
| MEC server | |
| Storage/RAM | 2048 Mb |
| Storage/HDD | 5 Gb |
| Processing/CPU | $\ni [0.7, 2.5]$ GHz |

The considered network for the simulation process is an IoT network with the structure introduced in Figure 2. Table 1 introduce the simulation parameters used to set up the simulation process. Each LiFi access point was connected with a fog node, while LiFi controllers were connected with MEC servers, one for each. The specifications of the considered fog nodes and MEC servers are introduced in Table 1.

A dataset of workloads of 100 heterogeneous tasks was constructed. Tasks correspond to the workload of real applications. Three categories of applications were assumed to be handled by the developed system. The first application category corresponds to applications with small workloads, such as medical applications, while the second category is introduced for applications with higher workloads, such as image-based applications. The third category is introduced for multimedia applications with a very high workload compared to the previous three workloads. Tasks are considered to arrive with a Poisson process of the rate indicated in Table 1.

The system was simulated for three scenarios, each corresponding to the use of a certain communication interface for IoT applications. The first scenario deploys only the LiFi interface with the hierarchal structure introduced in Figure 2. The second scenario uses a hybrid communication interface between LiFi and WiFi connections, while the third scenario deploys only WiFi as the short-range communication interface for end devices.

### 4.2. Simulation Results

Figure 5 provide the average offloading delay of tasks between IoT devices and edge units with the distance from the access points for the three considered scenarios. Using LiFi as the communication interface achieves the minimum offloading latency; however, the offloading latency increases with the distance from the sink, i.e., the access point. Our results indicate that the deployment of hybrid interfaces of LiFi and WiFi achieves higher latency efficiency than WiFi and LiFi for different positions from the access points. The hybrid scheme achieves a performance improvement of latency of an average of 62% compared to the WiFi scheme.

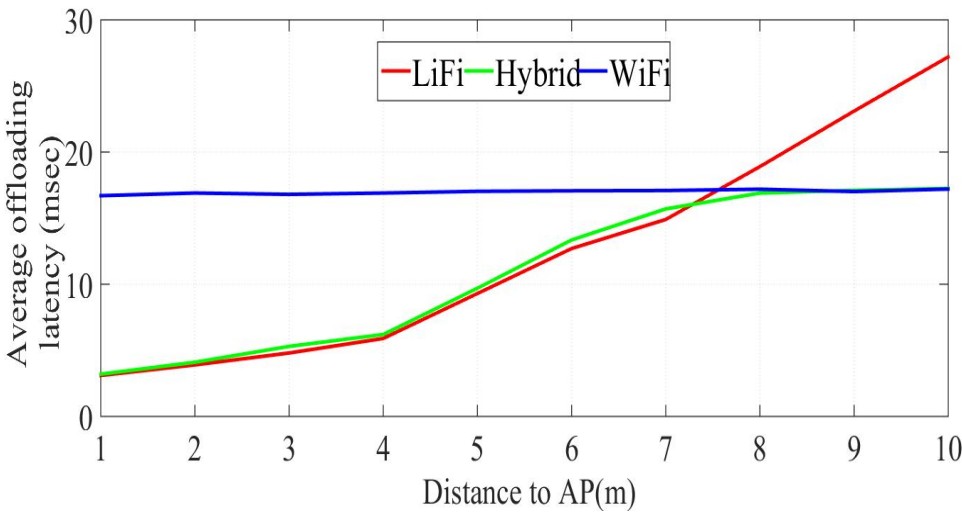

**Figure 5.** Average offloading latency of the three systems with the distance from the access point (AP).

Figure 6 introduce the communication overhead of the developed system for different numbers of deployed devices. Our results indicate that the communication overhead increases with the increase of the number of deployed devices; however, this undesirable effect differs according to the deployed communication interface. LiFi and the hybrid interfaces achieve higher efficiencies, i.e., less overhead, than traditional WiFi connection, especially with the increase in the number of deployed devices. The average percentage reduction of communication overhead when using a LiFi connection is 48%.

During the simulation procedure, four different systems were used to examine the impact of two tires of distributed edge computing. The fog-MEC system was the first system that was simulated. The second approach solely used edge computing units at the MEC level and no fog units were deployed. Distributed fog units with no MEC units were deployed in the third system, while the regular IoT network was used in the fourth.

System delay is shown in Figure 7 for five different simulated situations.

Simulated IoT networks include increasing numbers of deployed devices as they progress through the scenarios. This is a demonstration of the benefits of distributed edge computing. Introducing two tiers of edge computing units reduces the average time needed to perform computing operations. Compared to standard IoT networks with no edge computing units, the fog-MEC paradigm provides a performance boost of 67% in latency efficiency. The latency efficiency of IoT networks with only one level, i.e., fog nodes, improves by 46% when two layers of edge computing are included. Thus, the introduction

of another edge computing level reduces the communication latency by processing and handling IoT tasks near IoT devices with small communication distances.

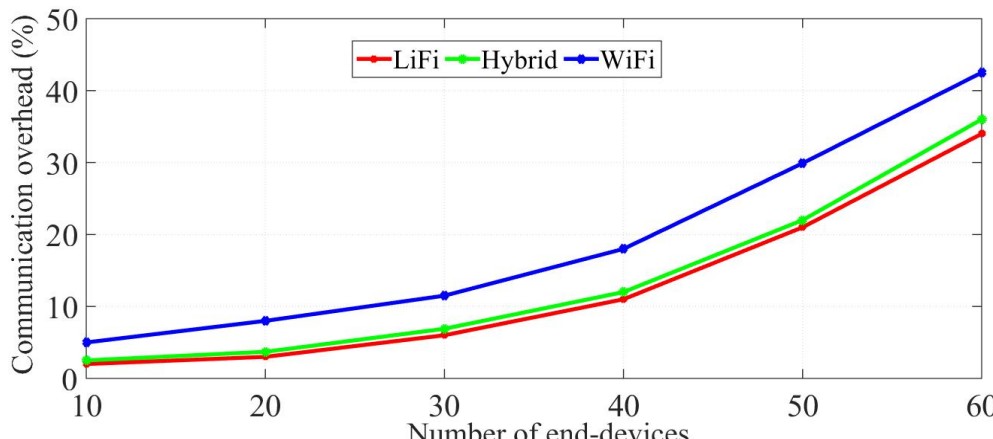

**Figure 6.** Communication overhead of the three considered systems for different numbers of end devices.

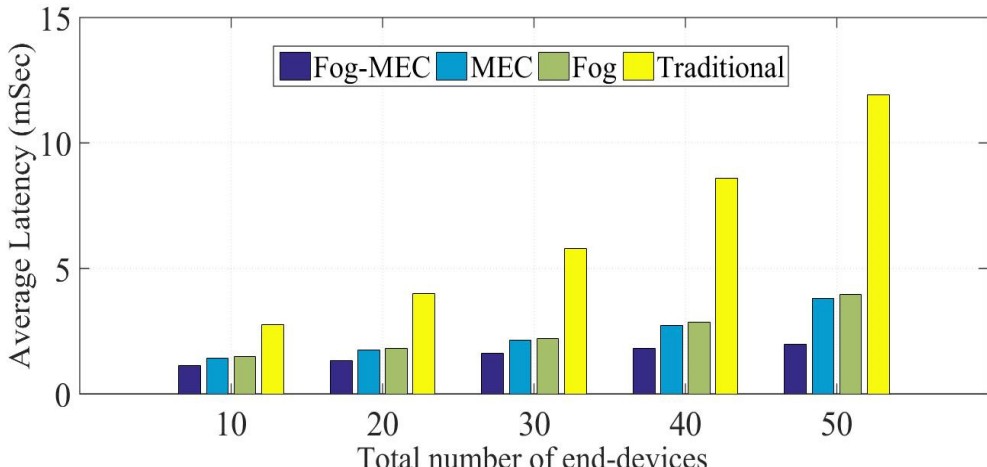

**Figure 7.** Average time of handling IoT tasks for the proposed fog-MEC model compared to other existing systems.

For each simulation scenario, the average number of blocked tasks is shown in Figure 8. With a limited number of end devices, the developed fog-MEC model reduces the number of blocked tasks to zero. With the increase in the number of end devices, the percentage of blocked tasks increases for all considered systems; however, the introduction of edge computing units reduces this challenge. The developed fog-MEC model reduces the percentage of blocked tasks by an average of 32% when deploying one level of edge computing and 67% when using standard IoT networks with no edge computing levels. The fog-MEC model, on the other hand, maximizes the use of edge computing units by increasing server usage. Figure 9 show that the created model outperforms other one-level systems by an average of 14% in resource usage efficiency.

The introduction of LiFi to IoT networks provides a novel way to achieve the required coverage, reliability, availability, and scalability. LiFi can be used as the communication interface for most indoor IoT applications with the ease of deployment. Deploying LiFi with WiFi as the communication interface for IoT applications achieves a 48% reduction of the communication overhead than existing traditional systems. Moreover, it reduces the offloading latency by an average of 62%.

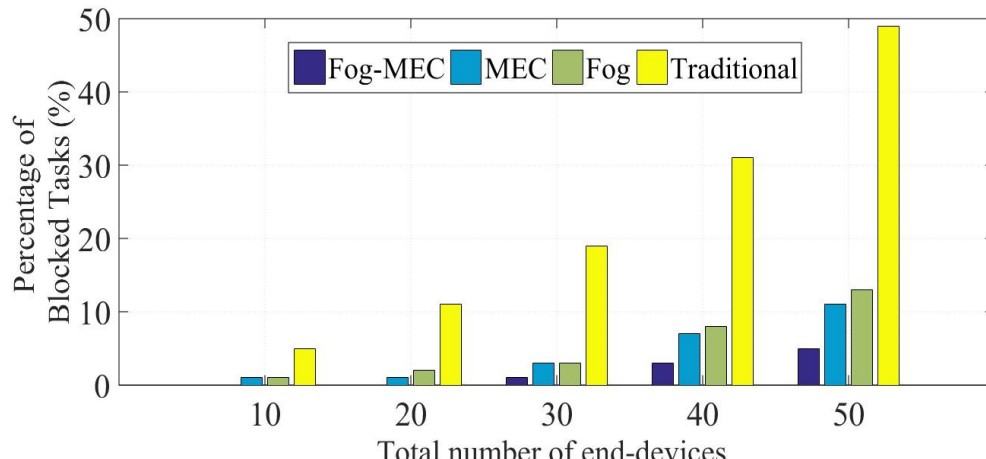

**Figure 8.** Percentage of blocked tasks for the proposed fog-MEC model compared to other existing systems.

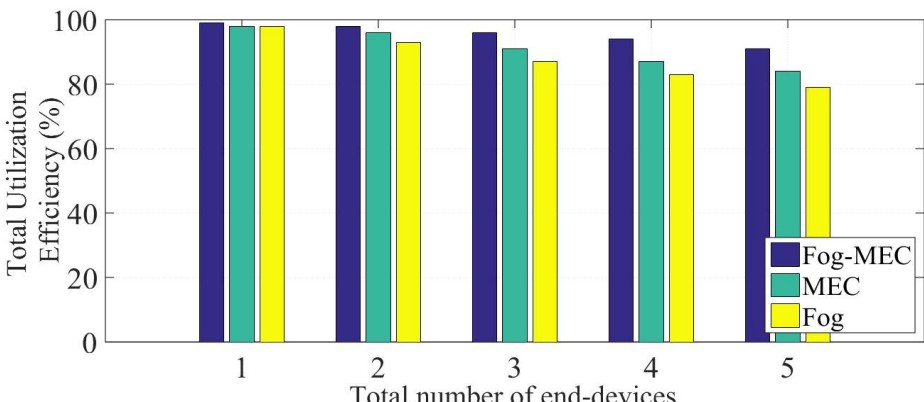

**Figure 9.** Total utilization efficiency of the proposed fog-MEC model compared to other existing systems.

Distributed edge computing is another paradigm that empowers the future of IoT networks. The proposed fog-MEC model archives many benefits to the considered LiFi-based IoT network that include reducing latency, higher utilization efficiency, and increased system availability. The fog-MEC model reduces the latency required to handle computing tasks by an average of 67% compared to traditional IoT systems, i.e., systems with no edge commuting units. Furthermore, it increases the efficiency of utilizing resources by an average of 14%.

## 5. Conclusions

The dramatic increase in the number of connected devices puts many constraints on network design in terms of the required scalability, availability, and reliability. The introduction of LiFi as the communication interface for indoor IoT applications is one way to overcome such challenges. The developed system provided an IoT network with LiFi communications arranged hierarchically. The proposed LiFi-based IoT network deploys a fog-MEC model, which is a novel structure of the edge computing paradigm. Introducing a two-level edge computing model achieves higher system availability by an average of 67% than the traditional IoT network structure. Moreover, the resource utilization increased by an average of 14%. Deploying LiFi with two-level edge computing reduces the communication overhead of IoT networks by an average of 48%.

**Author Contributions:** Conceptualization, A.A.A. and M.M.; methodology, A.A.A. and M.M.; software, A.A.A., A.Z. and M.M.; validation, M.M., A.M. and N.F.S.; formal analysis, A.A.A.; investigation, M.M. and N.F.S.; resources, A.Z. and N.F.S.; data curation, A.A.A. and M.M.; writing—original draft preparation, M.M., A.A.A. and N.F.S.; writing—review and editing, A.Z.; visualization, A.A.A.; supervision, A.Z.; project administration, A.M. and N.F.S.; funding acquisition, N.F.S. All authors have read and agreed to the published version of the manuscript.

**Funding:** Princess Nourah bint Abdulrahman University Researchers Supporting Project number (PNURSP2022R66), Princess Nourah bint Abdulrahman University, Riyadh, Saudi Arabia.

**Acknowledgments:** Princess Nourah bint Abdulrahman University Researchers Supporting Project number (PNURSP2022R66), Princess Nourah bint Abdulrahman University, Riyadh, Saudi Arabia.

**Conflicts of Interest:** The authors declare no conflict of interest.

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
