# Peer review of "Empowering the Internet of Things Using Light Communication and Distributed Edge Computing"

_electronics, doi:10.3390/electronics11091511_

Round 1

Reviewer 1 Report

The weakest part of the paper is that the contributions are very weak, it is not enough to present few figures of proposed idea and expect that to be accepted as journal paper. you should propose a complete model, with full theortical explanation to support you proposed system

-the introduction have excessive unnecessary citation, [3, 4]., [9-11], [13-14], [20, 21].,  [22, 23]. , [24] , most of these citation does not need more than 1 refererence, and in most cases there is no need for adding a reference at all. if you did that just to inflate the number of references, then you should remove most of these reference.

-consider adding the following related works:
[1] Chen, Chen, et al. "NOMA for energy-efficient LiFi-enabled bidirectional IoT communication." IEEE Transactions on Communications 69.3 (2021): 1693-1706.
[2] Naouri, Abdenacer, et al. "A Novel Framework for Mobile-Edge Computing by Optimizing Task Offloading." IEEE Internet of Things Journal 8.16 (2021): 13065-13076.

Reviewer 2 Report

- This paper designed a framework for IoT networks using LiFi and distributed edge computing scheme. It aims to enable the dense deployment, increase reliability and availability, and reduce the communication latency of IoT networks.

- This paper is well written and worthy to discuss.

- This paper described the contributions and method adequately. It makes a good description of the framework for IoT networks using LiFi and distributed edge computing scheme. Moreover, the author also provides the result and evaluation to demonstrate the contributions and the efficiency of the proposed model.

- However, there are some minor issues that can be considered to modify: the background and related works sections are presented quite long and rambling (4 pages), and they should be shortened and generalized.

- This paper is good enough to be accepted.

Author Response

We wish to thank the reviewers and the editor for their valuable comments and suggestions. We truly appreciate their suggestions and comments. Having detailed comments and recommendations helped us to significantly improve the manuscript.   In the following, we respond to the reviewers' comments and give the details of how they have been considered.

Reviewer #2:

  • This paper designed a framework for IoT networks using LiFi and distributed edge computing scheme. It aims to enable the dense deployment, increase reliability and availability, and reduce the communication latency of IoT networks.

This paper is well written and worthy to discuss.

This paper described the contributions and method adequately. It makes a good description of the framework for IoT networks using LiFi and distributed edge computing scheme. Moreover, the author also provides the result and evaluation to demonstrate the contributions and the efficiency of the proposed model.

However, there are some minor issues that can be considered to modify: the background and related works sections are presented quite long and rambling (4 pages), and they should be shortened and generalized.

This paper is good enough to be accepted.

  • We wish to thank the reviewer for the comment.
  • The section of background and related works was modified.

Reviewer 3 Report

To enable dense deployment, increase reliability and availability, and reduce the communication latency of IoT networks, this paper designs a general framework for IoT networks using LiFi and distributed edge computing techniques. The proposed framework both main types of distributed computing: multiple access edge computing and fog computing to achieve the requirements. The developed fog-MEC model is demonstrated in dense deployment scenarios, which is integrated to LiFi. It is evaluated for heterogeneous simulation scenarios, which is validated by the simulation results. The overall structure of this paper is reasonable and its expression is clear, but the demonstration method of the Related Work should be improved. It also lacks originality, and more technical details of the proposed LiFi-based IoT framework about Li-Communications and Fog MEC model are needed to be demonstrated. In a word, it needs a major revision before being accepted.

The authors should consider the following suggestions:

  • In Sec. 1, in Row 42 of Page 1, please add “as many as” before “the current”; in Row 53 of Page 2, please add “being” before “commercially”; in Row 74 of Page 2, please replace “distributing” with “distributed”. Please first introduce the relationship between distributed edge computing and distributed cloud in the paragraph “Limited computing resources of IoT … for edge computing”. please simplify the contribution statements of the work, and some are not really a contribution.
  • In Sec. 2, please replace “Background and Related Works” with “Background and Related Work”; it is suggested that the content of Background can be placed in Sec.1; in Sec.2.1, the advantages of introducing LiFi technique should be simplified, and they are verbose. It is better to cite the related references only. The demonstrated method of the related work should be improved, and the description of the references is too detailed, and their expression are not accurate. In Sec.2.2, there are the same problems as Sec.2.1.
  • In Sec.3, more technical details of the proposed LiFi-based IoT framework should be demonstrated. This proposed framework is not a novel idea, please give the difference from the edge computing-based IoT systems.
  • In Sec. 4, please first introduce the evaluation index. In Row 469 of Page 12, please explain “the best latency efficiency”. It is better to merge Sec. 4 with Sec.5.
  • The authors should examine the following studies, e.g., Security and trust issues in Fog computing: A survey; Recent Advances in Collaborative Scheduling of Computing Tasks in an Edge Computing Paradigm; Enhanced Intrusion Detection System for an EH IoT Architecture Using a Cooperative UAV Relay and Friendly UAV Jammer; Internet of Things for the Future of Smart Agriculture: A Comprehensive Survey of Emerging Technologies.
  • Many presentation issues need to be addressed, e.g., Line 3, Sec. 1, “between remote machines” => “among remote machines”. “Internet-connected devices went from 5 million to billions in just one year.” needs a reference (by perhaps indicating which specific year?).

Round 2

Reviewer 1 Report

N/A

Author Response

  • We wish to thank the reviewer.
  • We proofread the article.

Reviewer 3 Report

The authors have addressed most of my prior comments. yet they fail to discuss the suggested studies in the revised paper. In addition, the following studies should also be discussed: On the System Performance of Mobile Edge Computing in an Uplink NOMA WSN With a Multiantenna Access Point Over Nakagami-m Fading; An Edge Computing Paradigm for Massive IoT Connectivity Over High-Altitude Platform Networks; and IoT-enabled autonomous system collaboration for disaster-area management.

Second, Figs. 7-9 should be improved by not only using color but different texture inside each bar such that they can be seen in black/white printing.

Presentation needs improvement, e.g.,

Page 1, Line 39, "between machines " => "among machines"

Page 3, "In (Sec.2)" => "In Sec. 2". 

Author Response

Response Letter (point by point reply to reviewers' comments)

Empowering the Internet of Things Using Light Communication and Distributed Edge Computing

We wish to thank the reviewers and the editor for their valuable comments and suggestions. We truly appreciate their suggestions and comments. Having detailed comments and recommendations helped us to significantly improve the manuscript.   In the following, we respond to the reviewers' comments and give the details of how they have been considered.

Reviewer #3:

  • The authors have addressed most of my prior comments. yet they fail to discuss the suggested studies in the revised paper. In addition, the following studies should also be discussed: On the System Performance of Mobile Edge Computing in an Uplink NOMA WSN With a Multiantenna Access Point Over Nakagami-m Fading; An Edge Computing Paradigm for Massive IoT Connectivity Over High-Altitude Platform Networks; and IoT-enabled autonomous system collaboration for disaster-area management.

  • We wish to thank the reviewer for the comment.
  • The proposed work has been reconsidered in the revised manuscript.
  • We will consider the mentioned topics as a future work..

  • Presentation needs improvement, e.g.,

Page 1, Line 39, "between machines " => "among machines"

Page 3, "In (Sec.2)" => "In Sec. 2"..

  • We wish to thank the reviewer for the comment.
  • We proofread the article, and the mentioned errors were corrected.